# An Analytical Approach to Enhancing DNN Efficiency and Accuracy Using Approximate Multiplication

Salar Shakibhamedan [1]  Anice Jahanjoo [1]  Amin Aminifar [2]  Nima Amirafshar [2]  Nima TaheriNejad [2][1]
Axel Jantsch [1]

## Abstract

Achieving higher accuracy in Deep Neural Networks (DNNs) often reaches a plateau despite extensive training, retraining, and fine-tuning. This paper introduces an analytical study using approximate multipliers to investigate potential accuracy improvements. Leveraging the principles of the Information Bottleneck (IB) theory, we analyze the enhanced information and feature extraction capabilities provided by approximate multipliers. Through Information Plane (IP) analysis, we gain a detailed understanding of DNN behavior under this approach. Our analysis indicates that this technique can break through existing accuracy barriers while offering computational and energy efficiency benefits. Compared to traditional methods that are computationally intensive, our approach uses less demanding optimization techniques. Additionally, approximate multipliers contribute to reduced energy consumption during both the training and inference phases. Experimental results support the potential of this method, suggesting it is a promising direction for DNN optimization.

## 1. Introduction

Deep Neural Networks (DNNs) have achieved remarkable success across various applications, yet they often encounter a performance ceiling where traditional methods like retraining and fine-tuning fail to yield further accuracy improvements. This paper explores the innovative use of approximate multipliers as a form of approximate computing to break through these accuracy barriers. By integrating approximate multipliers into DNNs, we explore the potential for achieving improved accuracy levels. This pioneering approach leverages the IB theory to analyze and illustrate the improvements, providing a new paradigm in feature and information extraction. Additionally, our technique significantly reduces computational demands and energy consumption, offering an efficient alternative to conventional optimization techniques.

Despite the potential benefits of approximate multipliers, their application within the framework of IB theory remains unexplored. This study aims to fill this gap by presenting a comprehensive analysis of DNN behavior when approximate multipliers are employed. The following sections will discuss the role of approximate multipliers and the relevance of IB theory in understanding and enhancing DNN performance.

### 1.1. Approximate Multipliers

Approximate computing techniques, specifically approximate multipliers, have emerged as a promising approach to enhancing the efficiency and performance of DNNs. Traditional multipliers are resource-intensive, leading to high power consumption and latency, which are critical concerns in both the training and inference phases of DNNs. Approximate multipliers, by design, offer a trade-off between computational accuracy and resource utilization, resulting in significant reductions in power without severely compromising performance.

Several studies have explored different architectures of approximate multipliers tailored for DNN accelerators. For instance, adaptive fault-tolerant approximate multipliers have been proposed to mitigate soft errors and optimize hardware resources, achieving reliability levels close to exact multipliers while utilizing significantly less area and power (Taheri et al., 2024). Techniques like ScaleTRIM employ scalable truncation-based methods for integer multiplica-

---

[1]Institute of Computer Technology (ICT), TU Wien (Vienna University of Technology), Vienna, Austria [2]The Institute of Computer Engineering (ZITI), Heidelberg University, Heidelberg, Germany. Correspondence to: Salar Shakibhamedan <salar.shakibhamedan@tuwien.ac.at>, Anice Jahanjoo <anice.jahanjoo@tuwien.ac.at>, Amin Aminifar <amin.aminifar@uni-heidelberg.de>, Nima Amirafshar <nima.amirafshar@uni-heidelberg.de>, Nima TaheriNejad <nima@uni-heidelberg.de>, Axel Jantsch <axel.jantsch@tuwien.ac.at>.

Accepted to the Workshop on Advancing Neural Network Training at International Conference on Machine Learning (WANT@ICML 2024).

tion, leveraging curve fitting and error compensation to maintain accuracy while reducing hardware costs (Farahmand et al., 2023). Other innovative approaches include Dynamic Range Unbiased Multipliers (DRUM) (Hashemi et al., 2015) and Truncation-and-Rounding-Based Scalable Approximate Multipliers (TOSAM), which balance accuracy and efficiency in various DNN applications (Vahdat et al., 2019).

These innovations illustrate the potential of approximate multipliers to enhance DNN efficiency. However, their application to surpass accuracy limits in DNN training and inference remains underexplored. This paper seeks to extend the current understanding by demonstrating how approximate multipliers can break through the performance ceiling of DNNs and provide new insights into their inner workings through the lens of IB theory.

### 1.2. Information Bottleneck Theory

The IB theory provides a powerful framework for understanding and analyzing the behavior of DNNs (Tishby et al., 2000; Tishby & Zaslavsky, 2015). The IB principle focuses on the trade-off between compression and prediction, aiming to extract the most relevant information from the input data with respect to the output task (Hafez-Kolahi & Kasaei, 2019). By examining the mutual information between input, output, and hidden layers, the IB theory sheds light on the internal workings of DNNs and their generalization capabilities.

Previous research has demonstrated that DNN training involves distinct phases of fitting and compression, with the latter being critical for effective generalization (Hafez-Kolahi & Kasaei, 2019; Lorenzen et al., 2021). The IP, a visualization tool within the IB framework, plots the mutual information between layers and provides insights into the dynamics of information flow during training (Hafez-Kolahi & Kasaei, 2019; Goldfeld et al., 2018). This method has been used to reveal how information is processed and compressed in various network architectures, influencing their ability to generalize from training data to unseen data.

Despite its theoretical robustness, the application of IB theory to approximate computing, specifically with approximate multipliers, has not been investigated. This study aims to bridge this gap by employing IB analysis to elucidate the benefits of integrating approximate multipliers in DNNs, thereby uncovering new pathways to enhance accuracy and efficiency. Our approach involves using the IB framework to analyze the changes in information flow and compression when approximate multipliers are introduced, providing a detailed understanding of how these components influence DNN performance.

The remainder of this paper is organized as follows. Sec-

tion 2 presents our hypothesis, detailing the specific propositions we aim to test through our analysis. Section 3 provides a comprehensive background analysis, discussing the design and implementation of approximate multipliers and their integration with IB theory. Section 4 describes the experimental setup and discusses the results, highlighting the improvements in accuracy and power efficiency achieved through our approach. Finally, Section 5 concludes the paper and outlines potential directions for future research.

## 2. Hypothesis

In this section, we present our hypothesis, which forms the basis for our analysis and supports the results observed in our experiments.

We hypothesize that the inherent differences in the output results of approximate multipliers, which lead to variations in entropy, can significantly influence the information flow in DNNs. Traditional training processes, which consistently employ exact multiplication operations, may fail to achieve higher accuracy due to the limitations imposed by this uniform computational paradigm.

By contrast, the use of a set of approximate multipliers, each characterized by unique output behaviors and entropy levels, can introduce a heterogeneous computational paradigm within the DNN. We propose that this heterogeneous approach, wherein different layers of the network utilize different approximate multipliers, can enhance the information flow from input to output in a manner that is unattainable through conventional training and retraining processes.

To support our hypothesis, we use the information plane, a powerful tool that can visualize and illustrate the information flow in DNNs. The information plane has been utilized in many research works to study and demonstrate the dynamics of information processing within neural networks. By employing this tool, we aim to observe and analyze the impact of heterogeneous approximate multipliers on the information flow in DNNs. This visualization will help us understand how the variations in computational paradigms affect the network's ability to extract and process information, thereby providing empirical evidence for our hypothesis.

This approach, which can be considered a retraining method, achieves this improvement with a significantly lower number of computations compared to traditional retraining procedures. Our approach aims to efficiently enhance accuracy with reduced computational overhead, thus demonstrating the practical benefits and feasibility of our hypothesis.

In summary, our hypothesis is that the strategic use of heterogeneous approximate multipliers, with their distinct output entropies, can lead to a computational paradigm shift that

enables DNNs to achieve higher accuracy levels than those possible with traditional exact multiplication-based training processes. The information plane will be used to visualize these changes in information flow, and our proposed method will demonstrate the efficiency and effectiveness of this approach, thereby supporting our hypothesis through detailed empirical observations and practical validation.

## 3. Analytical Framework

### 3.1. Role of Approximate Multipliers in DNNs

Approximate multipliers are a type of approximate computing technique that introduces trade-offs between computational accuracy and speed, area or energy efficiency (or all three). These multipliers sacrifice computational accuracy by introducing errors in the results through the removal or simplification of certain hardware components (Amirafshar et al., 2023). While this can generate errors due to the lack of exact processing compared to precise multipliers, it can lead to lower delays in computations, reduced energy consumption (or increased power efficiency), potentially fewer components (depending on the technique used), and less area (less silicon costs).

Recent studies have demonstrated the utility of approximate multipliers in various machine learning and deep learning applications, which are inherently error-resilient (Shakibhamedan et al., 2024a; Damsgaard et al., 2024; TaheriNejad & Shakibhamedan, 2022). Many deep learning models can tolerate computational errors, allowing for more efficient operations with negligible accuracy drops that can be compensated or ignored depending on the application. Research indicates that using approximate multipliers instead of exact multipliers results in minor accuracy reductions, which are often acceptable within the context of deep learning tasks.

Quantization is another technique used to enhance efficiency and reduce energy consumption in deep learning and machine learning models (Liang et al., 2021). By reducing the bit width used in computations, quantization decreases the computational energy required and the memory footprint. One popular number representation in deep learning is INT8, which offers a suitable balance between dynamic range and computational efficiency. INT8 is supported by major deep learning frameworks such as TensorFlow (Abadi et al., 2015) and PyTorch (Paszke et al., 2019).

Recently, a family of quantized approximate multipliers for DNNs, known as Signed Carry Disregard Multipliers (SCDM8), was proposed (Shakibhamedan et al., 2024b). These multipliers introduce approximation by disregarding the carry value propagation from the lower bits to the higher bits, reducing the number of components compared to exact multipliers. This approach results in significant energy efficiency and power savings during multiplication operations,

*Table 1.* Energy Savings and Accuracy Degradation of Approximate Multipliers Compared to Exact Multipliers in Various DNN Architectures

| DNN | DATASET | ACCURACY (TOP-1) DEGRADATION | ENERGY SAVING |
|---|---|---|---|
| VGG16 | IMAGENET | 0.49% | 61% |
| RESNET152 | IMAGENET | 0.23% | 68% |
| MOBILENETV2 | IMAGENET | 0.10% | 45% |
| CONVVEXT-T | IMAGENET | 0.37% | 56% |
| LENET5-INSPIRED | MNIST | 0.07% | 51% |

albeit with some computational errors.

The SCDM8 family includes up to 100 variations of approximate multipliers, each representing different levels of approximation and error in the results. Experimental evaluations on well-known DNNs for image classification tasks, including VGG16/19, ResNet101/152, MobileNetV2, ConvNeXt, and a LeNet5-inspired CNN, demonstrated that 20 of these approximate multipliers provide satisfactory performance. These evaluations were conducted using Post-Training Quantization (PTQ) on pre-trained datasets.

The behavioral models of these multipliers, which simulate the output of the multiplication operations, are available at (Shakibhamedan et al., 2024b). According to the results of this study, these approximate multipliers achieve satisfactory accuracy for image classification tasks while delivering significant power savings. Some of the results, demonstrating energy savings compared to exact multipliers, are summarized in Table 1.

For our studies and experiments, we selected 20 approximate multipliers from the SCDM8 family, as identified in the original study. Using the naming convention SCDM8_XY (where X and Y represent the number of approximate bits), we chose the set SCDM8_XY (X:4–8, Y:1–4), which corresponds to 20 approximate multipliers labeled Approx_Mult 1-20 in our experiments.

To illustrate the computational errors introduced by approximate multipliers, we present the results of the exact multiplier, and 20 used approximate multipliers for input values in the range of [-10, 10]. These results are depicted in Appendix A.1, highlighting the trade-offs between accuracy and energy efficiency achieved by different levels of approximation.

The differences in outputs, which result from using different arithmetic operations, and their demonstrated performance in various DNNs, motivated us to study and analyze their performance in feature extraction from data. We also examined their effects on data encoding and decoding, as well as the information flow within DNNs. Our goal was to determine if these different computational paradigms could reveal information or phenomena that traditional computa-

tional paradigms cannot achieve.

## 3.2. Entropy Analysis

As the first step, we studied the entropy of the output of approximate multipliers and compared it to the output of the exact multiplier. Entropy, a concept introduced by Shannon (1948), quantifies information by measuring the uncertainty or the intrinsic amount of information associated with a variable. For a given discrete random variable $X$, with values coming from the set of possible values $X$ and density function $p(x)$, the entropy is given by:

$$H(X) = -\sum p(x) \log p(x) \qquad (1)$$

Due to the approximations in the multipliers' computations, the entropy of the outputs of approximate multipliers differs from that of exact multipliers. This difference directly affects feature extraction, information encoding, and information flow in DNNs. The results of this analysis are illustrated in Section 4.

As mentioned, these variations in computations motivated us to study their effects on the information flow within DNNs and examine how these computational paradigms can affect the accuracy of trained DNNs. Another question we aim to address is how the information flow is affected by heterogeneous computation flows when using different approximate multipliers in a DNN. Moreover, we study their effects on the 'fitting' and 'compressing' features in DNNs.

## 3.3. Information Theory Analysis in DNNs

For this purpose, we used the IB concept. The IB concept, proposed by Tishby & Zaslavsky (2015), studies and demonstrates information flow in DNNs using information theory.

In addition to entropy, another tool used in the IB concept is mutual information. Mutual information measures the mutual "knowledge" or "dependency" between two variables. Consider a second random variable $Y$, with the probability distribution $p(y)$ that takes values from the set of possible values $Y$. The joint probability distribution $p(x,y)$ represents the probability of $x$ and $y$ occurring together. Mutual information, expressed as $I(X;Y)$, measures how much information $X$ knows about $Y$ and is given by:

$$I(X;Y) = \sum_{x \in X} \sum_{y \in Y} p(x,y) \log \left( \frac{p(x,y)}{p(x)p(y)} \right) \qquad (2)$$

In the IB framework, each layer of a DNN is considered a random variable, including the first layer (input data) and the last layer (the output). For a DNN with $m$ hidden (middle) layers, the input layer $X$ and output layer $\hat{Y}$ are illustrated in Figure 1 According to the principles of the IB concept by

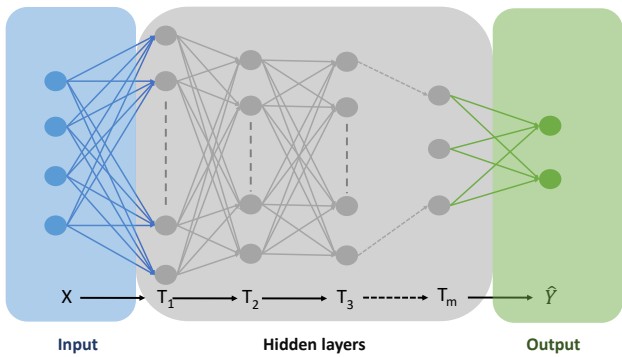

*Figure 1.* DNN schematic with input $X$, output $Y$, and hidden layers $T_1$ to $T_m$ for IB analysis.

Tishby & Zaslavsky (2015), if there are labels $Y$ associated with the input data $X$, a Markov chain can be applied to a DNN as follows:

$$Y \to X \to T_1 \to T_2 \to \ldots \to T_m \to \hat{Y} \qquad (3)$$

where $T_i$ denotes the $i$-th hidden layer, $X$ is the input, $Y$ is the corresponding label, and $\hat{Y}$ is the output of the DNN. According to (Tishby & Zaslavsky, 2015; Saxe et al., 2019), by applying the IB principle to a DNN, the hidden layers ($T_i$) are considered relevant components that "squeeze" the information from $X$ through a bottleneck in the form of the variable $T_i$. Essentially, $T_i$ (or generally $T$) is a compressed representation of $X$. The mutual information between $Y$ and $T$ ($I(Y;T)$) indicates the level of "informativeness" of $T$ about $Y$.

The IB method proposes that the optimal representation $T$ should maximize information about $Y$ while minimizing mutual information with $X$. In other words, simultaneously maximizing $I(T;Y)$ and minimizing $I(X;T)$ leads to better fitting and compression, respectively. This objective can be formulated as a Lagrangian:

$$L = I(Y;T) - \beta I(X;T) \qquad (4)$$

where $\beta$ is a regularization parameter.

By considering the applied IB to (3) by (Tishby & Zaslavsky, 2015) and according to the data processing inequality (DPI), the following inequalities hold for the information flow in DNNs:

$$I(Y;X) \geq I(Y;T_1) \geq \ldots \geq I(Y;T_m) \qquad (5)$$

$$I(X;X) \geq I(X;T_1) \geq \ldots \geq I(X;T_m) \qquad (6)$$

As shown by (5) and (6), the DPI imposes upper and lower bounds on $I(Y;T)$ and $I(X;T)$, which are considered in

our studies. According to the definitions in information theory and the IB concept, the terms of (4) can be reformulated using the following equations:

$$I(X;T) = H(T) - H(T|X) = H(T) \qquad (7)$$

$$I(T;Y) = I(Y;T) = H(T) - H(T|Y) \qquad (8)$$

where $H(T|Y)$ is the conditional entropy of $T$ given $Y$, and $H(T|X)$ is the conditional entropy of $T$ given $X$. Since $T$ is assumed to be a deterministic function of $X$, $H(T|X)$ equals zero.

By defining $I(X;T)$ and $I(Y;T)$, a visual representation called the information plane can be defined. In the information plane, the x-axis represents $I(X;T_i)$, and the y-axis represents $I(Y;T_i)$. This concept provides detailed insights into the information flow in each layer and the entire DNN. It is usually used in the training phase to visualize and study trends in information flow.

As discussed by Shwartz-Ziv & Tishby (2017) and Lorenzen et al. (2021) in several observations of training various datasets and using different activations (e.g., ReLU and Tanh), the fitting and compression phases have been observed. These phases, especially the compression phase, which is conjectured to contribute to good generalization, have been deeply studied in the literature (Yu et al., 2020; Kirsch et al., 2020; Shwartz-Ziv & Alemi, 2020; Jónsson et al., 2020; Darlow & Storkey, 2020; Goldfeld et al., 2018).

### 3.4. Improving DNNs with Heterogeneous Computational Paradigms

Our approach harnesses the different computational paradigms and output entropies of approximate multipliers to examine new possibilities for feature extraction and information flow. These paradigms enable us to reach certain points on the information plane that are unattainable with traditional computational paradigms, such as conventional training methods.

We observed that the output entropy of each approximate multiplier is distinct from one another and from the entropy of exact multipliers. This diversity provides a range of computational paradigms. Our novel approach involves using a different approximate multiplier for each layer in a DNN, creating a heterogeneous computational environment. This heterogeneity, arising from varying levels of multiplication entropy, positively impacts the information flow on the information plane, leading to better fitting (accuracy) and compression. To identify suitable configurations for such a heterogeneous implementation, we utilized a genetic algorithm, as detailed in Section 4. Moreover, as previously

mentioned, the power consumption of approximate multipliers is lower compared to exact multipliers. Therefore, any layer-wise combination of approximate multipliers will have less power consumption compared to a fully quantized INT8 DNN.

In summary, by using a heterogeneous information flow with different multipliers, we achieved better accuracy (fitting) and compression, along with improved power efficiency. This can be considered a novel technique for retraining, which offers significant benefits in terms of accuracy, compression, and energy efficiency during both training and inference.

## 4. Experimental Setup and Results

In this section, we present and illustrate the results of our study and the conducted experiments. As mentioned in the previous section, we used 20 quantized (INT8) approximate multipliers from (Shakibhamedan et al., 2024b) for our experiments. Due to the applied approximations, the proposed approximate multipliers consume less energy for performing multiplications, thus achieving power efficiency. The power efficiency of the used approximate multipliers is illustrated in Appendix A.2. The power efficiency is calculated in comparison to exact multipliers. The ACE-CNN (Shakibhamedan et al., 2024b) implemented approximate multipliers homogeneously, meaning they used one of the approximate multipliers for all layers in the DNNs.

For our experiments, we select the LeNet5-inspired CNN, which was proposed and used in (Shakibhamedan et al., 2024b), as our case study. This CNN is trained on the MNIST dataset (Deng, 2012). We also used the MNIST dataset in our experiments. The architecture of the used CNN is illustrated in Table 2.

*Table 2.* Architecture of the LeNet5-inspired CNN for MNIST dataset experiments ($T$ stands as the batch size).

| Layer Type (Name) | Filters | Output Dimension |
|---|---|---|
| Conv2D (Conv Layer_1) | [1x1x64] | $T$x28x28x64 |
| Activat (Relu) | - | $T$x28x28x64 |
| Conv2D (Conv Layer_2) | [1x1x32] | $T$x28x28x32 |
| Activat (Relu) | - | $T$x28x28x32 |
| Conv2D (Conv Layer_3) | [1x1x16] | $T$x28x28x16 |
| Activat (Relu) | - | $T$x28x28x16 |
| Conv2D (Conv Layer_4) | [3x3x8] | $T$x26x26x8 |
| Activat (Relu) | - | $T$x26x26x8 |
| Conv2D (Conv Layer_5) | [3x3x4] | $T$x24x24x4 |
| Activat (Relu) | - | $T$x24x24x4 |
| Flatten | - | - |
| FC (FC Layer_1) | [2304x128] | $T$x128 |
| FC (FC Layer_2) | [128x64] | $T$x64 |
| FC (FC Layer_3) | [64x10] | $T$x10 |

As mentioned before, the entropy of the approximate multipliers' output is different, providing us with various compu-

tational paradigms. The entropy of exact and approximate multipliers is presented in Appendix A.3. The entropy is calculated by considering all possible $256 \times 256$ inputs.

As the first step, we trained the mentioned CNN proposed by Shakibhamedan et al. (2024b) from scratch (in float32 number representation) to reach the reported accuracy. In our experiments, we trained the CNN with the MNIST dataset (training data of MNIST). The batch size was 1024, the loss function was set to cross-entropy, and the Adam optimizer (with a learning rate of $10^{-4}$) was used (with 5000 epochs) for optimization. The output of each layer during training was saved for calculating the needed mutual information $I(X;T)$ and $I(Y;T)$. After training, the information plane of this training procedure was computed and illustrated in Figure 2.

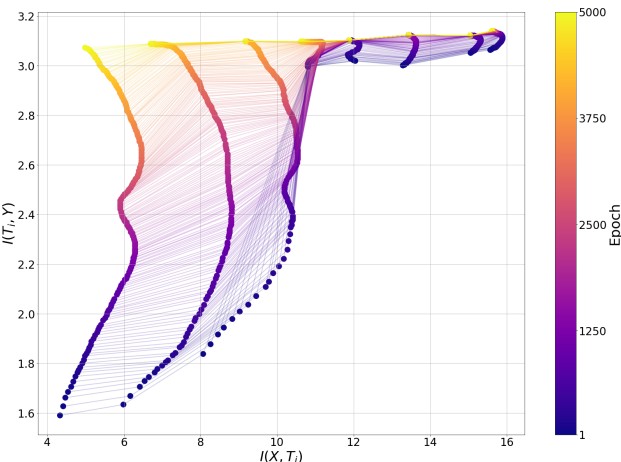

*Figure 2.* Information plane of LeNet5-inspired CNN training. The x-axis shows $I(X;T_i)$ and the y-axis shows $I(Y;T_i)$. The right-most curve is the first layer, and the leftmost curve is the last layer, in sequential order.

As shown, the information curves of convolutional layers have different structures, and various trends in fitting and compressing can be observed. Similar to the ACE-CNN research work (Shakibhamedan et al., 2024b), we applied PTQ to the weights. However, we did not limit applying PTQ just after training. Instead, we applied PTQ after each epoch to calculate the output of each layer in a quantized manner. This means that the saved output of each layer was quantized throughout the training process, ensuring that all mutual information calculations were performed on quantized data.

As mentioned in (Lorenzen et al., 2021), computing MI in quantized neural networks is exact and has higher accuracy compared to calculating MI for non-quantized neural networks. We used the proposed tool in (Lorenzen et al., 2021) for calculating the values of MI.

The information values of the last epoch are illustrated in Figure 3. Next, to study the effects of using approximate

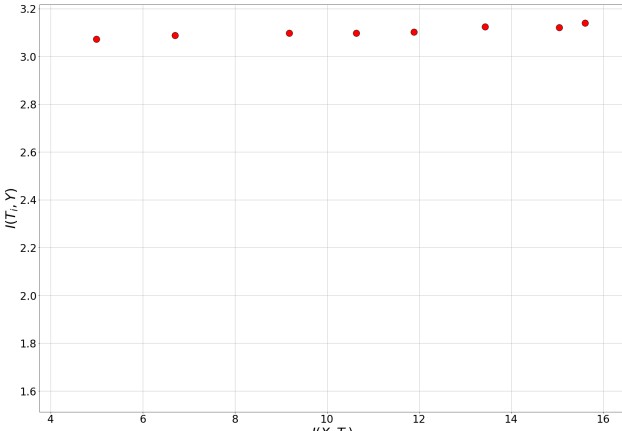

*Figure 3.* Information values of the last epoch for each layer of the LeNet5-inspired CNN. The plot shows the mutual information between the input $X$ and the hidden layers $T_i$ $I(X;T_i)$ on the x-axis, and the mutual information between the output $Y$ and the hidden layers $T_i$ $I(Y;T_i)$ on the y-axis, illustrating the information flow and the final fitting and compression states of the network.

multipliers on the information values of the last epoch in Figure 3, we replaced the exact multiplier in each layer with the proposed approximate multipliers. The achieved accuracy, power efficiency (due to using approximate multipliers), and information values are reported in Appendix A.4 and Figure 4, respectively.

As shown, there are some drifts in the information values compared to the values from Figure 2. Using approximate multipliers has led to more compression (reducing the values of $I(X;T)$), which shows the effects of using different computational paradigms. On the other hand, a drift in the values of $I(Y;T)$ (fitting procedure) can also be observed, which occurs due to the computational errors and accuracy reduction reported in Appendix A.4.

We used these findings to conduct our ultimate goal of implementing layer-wise heterogeneous approximate multipliers to find the point on the information plane that not only has more compression (similar to what happened in homogeneous implementation) but also achieves more accuracy (increasing the values of $I(Y;T)$ in the fitting phase). For this experiment, we used a genetic algorithm to achieve more computational efficiency compared to retraining (more details are discussed in Appendix C).

For implementing the genetic algorithm, we set the input number to 8 (as we have 8 layers in our CNN), the population size to 50, and the number of generations to 25. The mutation rate was set to 0.1. The details of the implemented

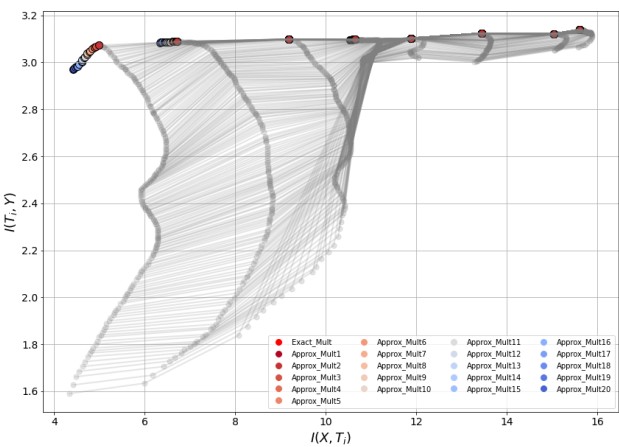

*Figure 4.* Information values of the last epoch after replacing exact multipliers with approximate multipliers in each layer of the LeNet5-inspired CNN. The x-axis shows $I(X; T_i)$ and the y-axis shows $I(Y; T_i)$, highlighting the changes in information flow, fitting, and compression due to the use of approximate multipliers.

*Table 3.* Energy savings and accuracy improvement of heterogeneous approximate multipliers implementation compared to the exact multiplier with an accuracy of 99.25%.

| APPROX_MULT COMBINATION | ACCURACY (TOP-1) | ENERGY SAVING |
|---|---|---|
| 19,10,6,3,10,6,12,10 | 99.29% | 56% |
| 18,5,14,10,3,8,6,19 | 99.36% | 57% |
| 18,5,6,13,7,3,1,14 | 99.41% | 50% |
| 12,8,14,2,5,9,15,2 | 99.45% | 61% |

genetic algorithm are described in Algorithm 1.

By conducting the genetic algorithm (which can be considered a form of re-training), we found some combinations of approximate multipliers that not only provide more compression but also deliver higher accuracy. These combinations allowed us to reach points on the information plane that could not be reached using traditional training procedures and exact INT8 multipliers. The achieved accuracies and the points reached on the information plane are illustrated in Table 3 and Figure 5, respectively. Moreover, the energy efficiency of inference, due to using heterogeneous approximate multipliers during the inference phase, is also reported in Table 3 (details are discussed in Appendix A.4.1). In summary, our experiments demonstrated that by using a heterogeneous information flow with different approximate multipliers, we can achieve better accuracy (fitting) and compression, along with improved power efficiency. Such a 'retraining' can be performed using simple algorithms such as genetic algorithm. This novel method for retraining

**Algorithm 1** Details of implemented genetic algorithm

**Input:** Initial Approx_Mult for: $Conv\ Lyr\_1$, $Conv\ Lyr\_2$, $Conv\ Lyr\_3$, $Conv\ Lyr\_4$, $Conv\ Lyr\_5$, $FC\ Lyr\_1$, $FC\ Lyr\_2$, $FC\ Lyr\_3$
**Set:**
Generation_Num = 50
Population_Size = 50
Mutation_rate = 0.1
Input_range: Approx_Mult 1:20
Desired_output (Accuracy) = 1
Create the initial population
**Output:** Desired Approx_Mult combination:
CL1, CL2, CL3, CL4, CL5, FC1, FC2, FC3
**repeat**
    1. Evaluate the fitness of each individual
    2. Select parents for reproduction
    3. Create the next generation through crossover and mutation
    4. Replace the old population with the offspring
**until** Number of Generations
Find the best individual in the final population
Return the desired results.

offers significant benefits in terms of accuracy, compression, and efficiency during both training and inference. The results highlight the potential of approximate multipliers to surpass the prediction performance limits of traditional methods while reducing computational costs and energy consumption.

## 5. Conclusion and Discussion

In this section, we summarize our findings and discuss the implications of using approximate multipliers in DNNs, focusing on accuracy improvements, energy efficiency, and potential future research directions.

The results of our study demonstrate that approximate multipliers can enhance the performance (accuracy) and efficiency of DNNs. By incorporating these multipliers, we were able to achieve higher accuracy levels that surpass the traditional limits encountered with conventional training methods. This was demonstrated through the novel application of the IB theory, which provided a comprehensive framework to understand the effects of using approximate multipliers and optimizing the information flow within the network.

Our experiments with the LeNet5-inspired CNN on the MNIST dataset highlighted several key advantages of using approximate multipliers:

1. **Improved Accuracy**: Using a heterogeneous combination of approximate multipliers for each layer, re-

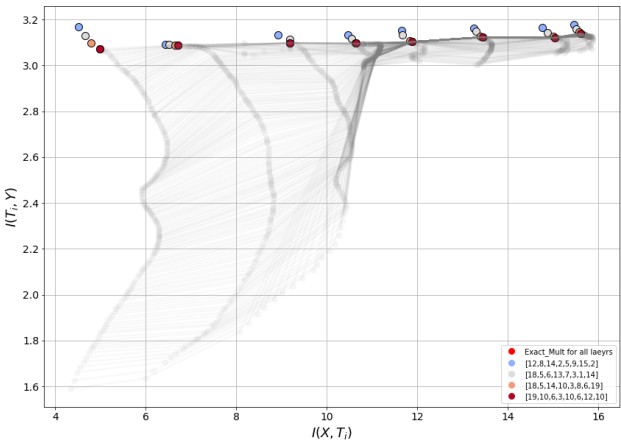

*Figure 5.* Information plane of the LeNet5-inspired CNN after implementing layer-wise heterogeneous approximate multipliers. The x-axis represents $I(X; T_i)$ and the y-axis represents $I(Y; T_i)$. This figure highlights the points on the information plane achieved through genetic algorithm optimization, showing enhanced compression and fitting compared to traditional training methods.

sulted in improved fitting and compression, confirming our hypothesis. This led to higher accuracy levels, as shown by the enhanced mutual information values in the information plane.

2. **Energy Efficiency**: The use of approximate multipliers resulted in significant power savings during both the training and inference phases. This efficiency is particularly beneficial for edge and IoT applications where energy consumption is a critical factor.

3. **New Retraining Approach:** We proposed a retraining approach that offers multiple benefits. First, this method is significantly more efficient, as discussed in detail in Appendix C, where we compare normal retraining (training) with a simpler algorithmic approach. This novel approach leverages a new computational paradigm that is distinct from those used in conventional training and retraining methods. It benefits from mathematical principles and insights from information theory-based analysis, which are not utilized in current methods.

4. **Information Plane Discovering**: Using the information plane as a tool for visualizing what is happening within the DNNs, we demonstrated that our efficient proposed training method can reach points on the information plane that other methods cannot achieve. This highlights the unique advantages of our approach in terms of information flow and network optimization.

### 5.1. Future Research Directions

While our study has established the potential of approximate multipliers in enhancing DNN performance, several avenues for future research remain:

1. **Broader Application Scope**: Extending the application of approximate multipliers to other types of neural networks and datasets could provide further validation of their effectiveness and versatility.

2. **From-Scratch Training**: One of the future directions we plan to explore is using this method for from-scratch training. This could provide further insights into the fundamental advantages of using approximate multipliers throughout the entire training process.

3. **Optimization Algorithms**: Exploring other optimization techniques beyond genetic algorithms could yield even more efficient methods for selecting approximate multipliers.

4. **Theoretical Foundations**: Further theoretical exploration of the relationship between approximation levels, entropy, and mutual information could deepen our understanding of the fundamental principles governing DNN behavior. One potential avenue is to incorporate such analysis results as feedback for selecting the approximate multipliers. For instance, introducing a term based on a parameter from this analysis into the fitness function used in our optimization algorithm could enhance the selection process and overall performance.

In conclusion, the use of approximate multipliers in conjunction with the IB theory represents a promising direction for advancing the capabilities of DNNs. Our findings underscore the potential for achieving superior accuracy and energy efficiency, paving the way for more robust and sustainable AI applications.

### Acknowledgment

The authors gratefully acknowledge funding from European Union's Horizon 2020 Research and Innovation Programme under the Marie Skłodowska Curie grant agreement No. 956090 (APROPOS: Approximate Computing for Power and Energy Optimisation, http://www.apropos.eu/ ).

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

# A. Appendix

## A.1. Visualization of Values for All Approximate Multipliers and Exact Multipliers

In this appendix, we provide a detailed visualization of the outputs for all 20 approximate multipliers and the exact multiplier when the input values range from $[-10, 10]$. This visualization helps to illustrate the differences in computational results due to the approximations and to highlight the trade-offs between accuracy and energy efficiency achieved by different levels of approximation. To visualize the output, we used a color spectrum from red to green (from the lowest value to the highest value).

The following figures present the output results for the exact multiplier and each of the 20 approximate multipliers (labeled Approx_Mult1 to Approx_Mult20). Each figure plots the output of the multiplication for input values in the range of $[-10, 10]$.

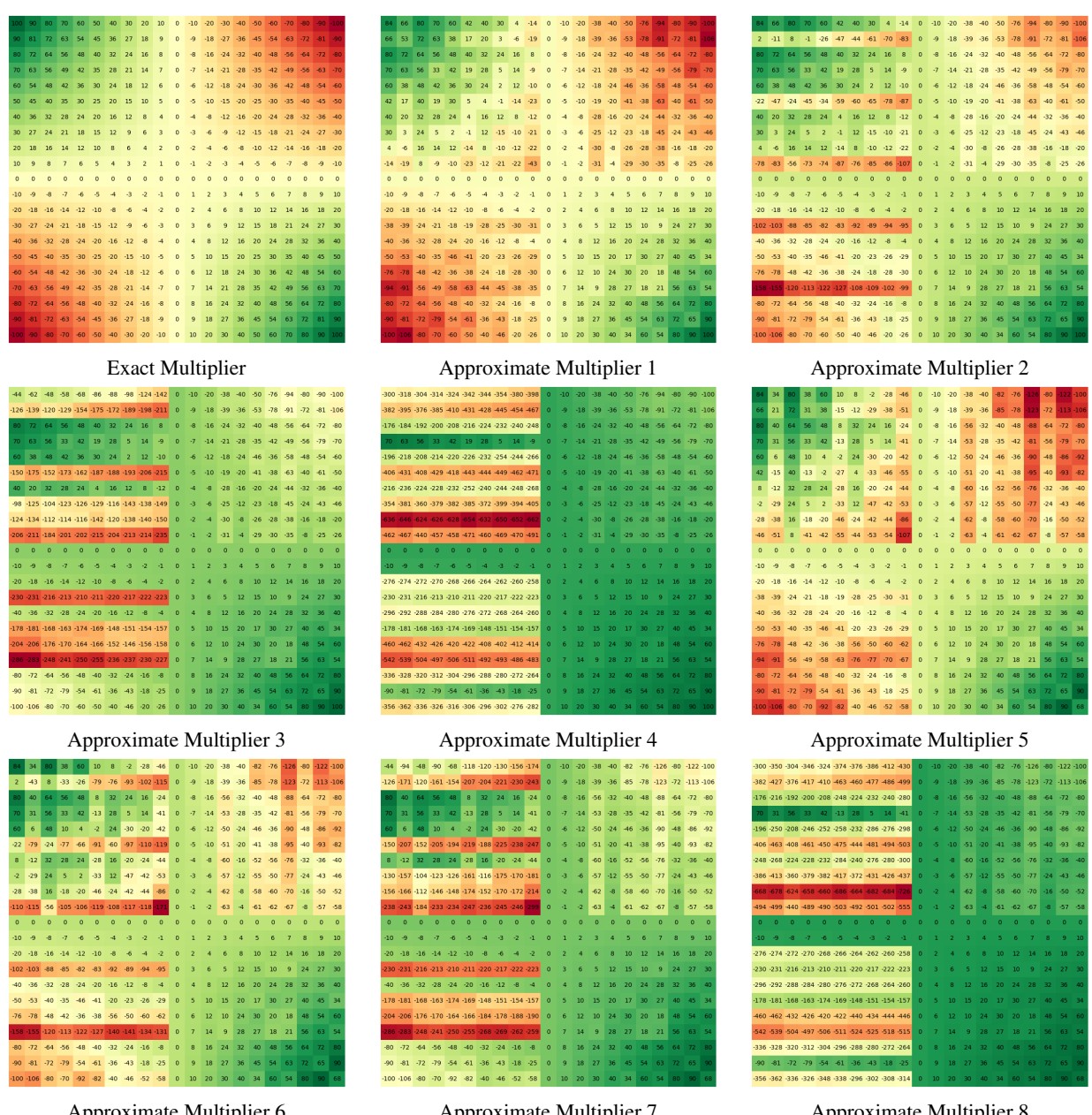

Exact Multiplier      Approximate Multiplier 1      Approximate Multiplier 2

Approximate Multiplier 3      Approximate Multiplier 4      Approximate Multiplier 5

Approximate Multiplier 6      Approximate Multiplier 7      Approximate Multiplier 8

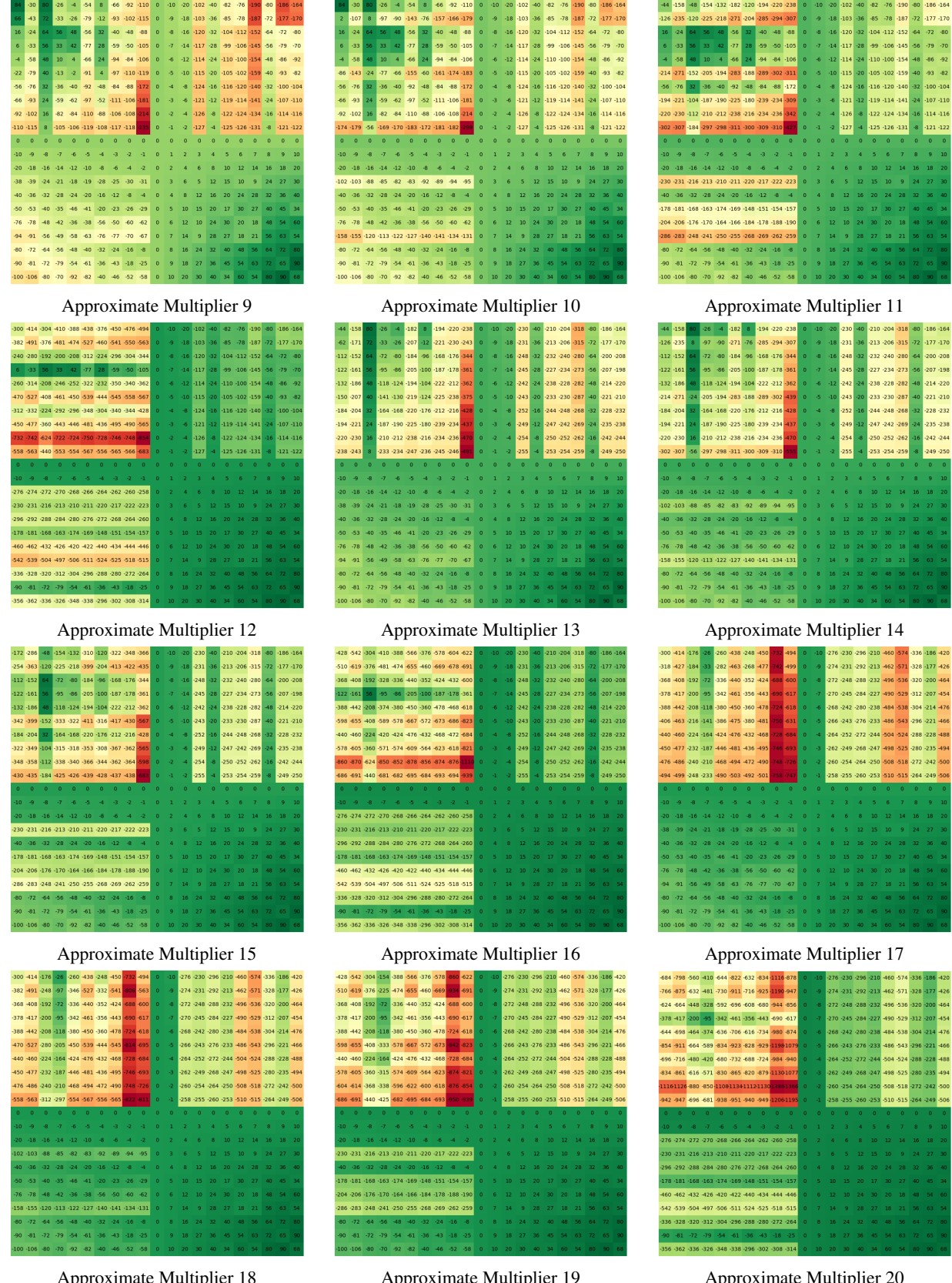

*Figure 6.* Output of exact multiplier and approximate multipliers (Approx_Mult1 to Approx_Mult20) for input values in the range [−10, 10]. Each subplot uses a color spectrum from red to green to represent the output values from lowest to highest.

The visualizations provided in these figures clearly demonstrate the computational differences introduced by each approximate multiplier compared to the exact multiplier. These differences are crucial for understanding the trade-offs between energy efficiency and accuracy in DNN computations.

## A.2. Power Efficiency of Approximate Multipliers

In this section, we provide information about the power efficiency of the approximate multipliers used in our experiments. The power efficiency is measured by the power usage coefficient, which compares the energy consumption of each approximate multiplier to that of the exact multiplier. A lower power usage coefficient indicates higher energy efficiency.

| Type of Multiplier | Power Usage Coefficient |
|---|---|
| Exact Multiplier | 1.00 |
| Approximate Multiplier 1 | 0.78 |
| Approximate Multiplier 2 | 0.76 |
| Approximate Multiplier 3 | 0.74 |
| Approximate Multiplier 4 | 0.69 |
| Approximate Multiplier 5 | 0.69 |
| Approximate Multiplier 6 | 0.64 |
| Approximate Multiplier 7 | 0.61 |
| Approximate Multiplier 8 | 0.58 |
| Approximate Multiplier 9 | 0.66 |
| Approximate Multiplier 10 | 0.61 |
| Approximate Multiplier 11 | 0.58 |
| Approximate Multiplier 12 | 0.55 |
| Approximate Multiplier 13 | 0.63 |
| Approximate Multiplier 14 | 0.54 |
| Approximate Multiplier 15 | 0.49 |
| Approximate Multiplier 16 | 0.43 |
| Approximate Multiplier 17 | 0.58 |
| Approximate Multiplier 18 | 0.52 |
| Approximate Multiplier 19 | 0.46 |
| Approximate Multiplier 20 | 0.40 |

*Table 4.* Power Usage Coefficients for Exact and Approximate Multipliers

## A.3. Entropy of Outputs from Exact and Approximate Multipliers

In this section, we provide information about the entropy of the outputs from the exact and approximate multipliers. Entropy, a concept introduced by Shannon, is a measure of uncertainty or the amount of information inherent in a variable. The entropy of the output reflects the variability and unpredictability of the results produced by the multipliers. Higher entropy indicates more variability in the output.

| Type of Multiplier | Output Entropy | Type of Multiplier | Output Entropy | Type of Multiplier | Output Entropy |
|---|---|---|---|---|---|
| Exact Multiplier | 12.856 | Approximate Multiplier 7 | 13.458 | Approximate Multiplier 14 | 13.525 |
| Approximate Multiplier 1 | 13.183 | Approximate Multiplier 8 | 13.600 | Approximate Multiplier 15 | 13.502 |
| Approximate Multiplier 2 | 13.261 | Approximate Multiplier 9 | 13.438 | Approximate Multiplier 16 | 13.637 |
| Approximate Multiplier 3 | 13.372 | Approximate Multiplier 10 | 13.436 | Approximate Multiplier 17 | 13.625 |
| Approximate Multiplier 4 | 13.580 | Approximate Multiplier 11 | 13.516 | Approximate Multiplier 18 | 13.633 |
| Approximate Multiplier 5 | 13.325 | Approximate Multiplier 12 | 13.632 | Approximate Multiplier 19 | 13.634 |
| Approximate Multiplier 6 | 13.379 | Approximate Multiplier 13 | 13.513 | Approximate Multiplier 20 | 13.641 |

*Table 5.* Output Entropy for Exact and Approximate Multipliers

### A.4. Benefits of Replacing Exact Multiplier with Approximate Multipliers for Inference

In this section, we present the benefits of replacing the exact multiplier with approximate multipliers during the inference phase after training. The use of approximate multipliers can lead to significant energy savings while maintaining high accuracy. The table below shows the accuracy (TOP-1) and energy savings achieved by using various approximate multipliers compared to the exact multiplier.

| Type of Multiplier | Accuracy (TOP-1) | Energy Saving (%) |
|---|---|---|
| Exact Multiplier | 99.25% | 0% |
| Approximate Multiplier 1 | 99.25% | 28% |
| Approximate Multiplier 2 | 99.24% | 32% |
| Approximate Multiplier 3 | 99.04% | 36% |
| Approximate Multiplier 4 | 98.27% | 44% |
| Approximate Multiplier 5 | 99.20% | 45% |
| Approximate Multiplier 6 | 99.18% | 56% |
| Approximate Multiplier 7 | 98.91% | 63% |
| Approximate Multiplier 8 | 98.72% | 72% |
| Approximate Multiplier 9 | 99.12% | 52% |
| Approximate Multiplier 10 | 99.21% | 64% |
| Approximate Multiplier 11 | 98.61% | 71% |
| Approximate Multiplier 12 | 98.44% | 82% |
| Approximate Multiplier 13 | 98.70% | 60% |
| Approximate Multiplier 14 | 98.62% | 84% |
| Approximate Multiplier 15 | 98.98% | 106% |
| Approximate Multiplier 16 | 97.24% | 130% |
| Approximate Multiplier 17 | 89.06% | 71% |
| Approximate Multiplier 18 | 94.60% | 94% |
| Approximate Multiplier 19 | 94.57% | 118% |
| Approximate Multiplier 20 | 94.47% | 151% |

*Table 6.* Accuracy (TOP-1) and Energy Savings for Exact and Approximate Multipliers During Inference

This table highlights the trade-offs between accuracy and energy savings when using approximate multipliers for inference. The results show that it is possible to achieve significant energy savings with only a small reduction in accuracy. In some cases, approximate multipliers can even provide energy savings of over 100%, demonstrating their potential for improving the energy efficiency of DNNs without substantial sacrifices in performance.

### A.4.1. ENERGY SAVING CALCULATION

According to the architecture of the used DNN (Table 2), the number of required multiplications is calculated for each layer. The number of multiplications in each layer is also detailed in Appendix C. To define energy saving, we consider the power usage of the exact multiplier as 1, and the power usage of the approximate multipliers as reported in Table 4 in Appendix A.2. The energy saving is calculated as the ratio of the power usage of the exact multiplier to the power usage of the approximate multipliers.

Table 7 below provides a breakdown of the required multiplications for each layer in the DNN architecture. To calculate the energy saving values of Table 3 when we use the approximate multipliers heterogeneously for the layers in the DNN, we consider the energy usage of the exact multiplier as $E_x$ and the energy usage of the used approximate multiplier for the $i$-th layer as $E_{p_i}$. The energy saving is calculated using the following formula:

$$\text{Energy Saving} = \left( \frac{50176 \cdot \left( \frac{E_x}{E_{p1}} \right) + 1605632 \cdot \left( \frac{E_x}{E_{p2}} \right) + 401408 \cdot \left( \frac{E_x}{E_{p3}} \right) + 778752 \cdot \left( \frac{E_x}{E_{p4}} \right) + 165888 \cdot \left( \frac{E_x}{E_{p5}} \right) + 294912 \cdot \left( \frac{E_x}{E_{p6}} \right) + 8192 \cdot \left( \frac{E_x}{E_{p7}} \right) + 640 \cdot \left( \frac{E_x}{E_{p8}} \right)}{3586592} - 1 \right) \times 100$$

For each combination of heterogeneous approximate multipliers, the values of $E_x$ and $E_{p_i}$ are the corresponding values from Table 4.

*Table 7.* Energy Saving Calculation for CNN Architecture

| LAYER TYPE (NAME) | FILTERS | OUTPUT DIMENSION | NUMBER OF MULTIPLICATIONS |
|---|---|---|---|
| CONV2D (CONV LAYER_1) | [1x1x64] | $T$x28x28x64 | 50176 |
| ACTIVAT (RELU) | - | $T$x28x28x64 | - |
| CONV2D (CONV LAYER_2) | [1x1x32] | $T$x28x28x32 | 1605632 |
| ACTIVAT (RELU) | - | $T$x28x28x32 | - |
| CONV2D (CONV LAYER_3) | [1x1x16] | $T$x28x28x16 | 401408 |
| ACTIVAT (RELU) | - | $T$x28x28x16 | - |
| CONV2D (CONV LAYER_4) | [3x3x8] | $T$x26x26x8 | 778752 |
| ACTIVAT (RELU) | - | $T$x26x26x8 | - |
| CONV2D (CONV LAYER_5) | [3x3x4] | $T$x24x24x4 | 165888 |
| ACTIVAT (RELU) | - | $T$x24x24x4 | - |
| FLATTEN | - | - | - |
| FC (FC LAYER_1) | [2304x128] | $T$x128 | 294912 |
| FC (FC LAYER_2) | [128x64] | $T$x64 | 8192 |
| FC (FC LAYER_3) | [64x10] | $T$x10 | 640 |
| TOTAL | - | - | **3586592** |

In summary, the energy saving is calculated by taking the weighted average of the energy savings for each layer, considering the number of multiplications in each layer, and then multiplying by 100 to express it as a percentage.

## B. Appendix-Training Parameters Evaluation

During the training phase, we experimented with various parameters over 5,000 epochs to determine the optimal settings for our study. We evaluated different combinations of batch sizes, learning rates, and other hyperparameters. Despite extensive experimentation, the parameters detailed in the Experiments chapter were chosen as they consistently yielded the best results in terms of accuracy and efficiency.

The parameters we tested include:

- **Batch sizes:** Ranging from 256 to 4096

- **Learning rates:** Ranging from $10^{-5}$ to $10^{-3}$

- **Normalization techniques:** Including batch normalization

Batch normalization and dropout were used in all training configurations, including the chosen one.

After thorough evaluation, the combination of a batch size of 1024, a learning rate of $10^{-4}$, and the Adam optimizer with a cross-entropy loss function was found to be the most effective. These parameters provided the highest accuracy and stability throughout the training process, as detailed in the Experiments chapter.

To ensure a fair comparison, we used 10,000 images from the dataset as the validation dataset in the training procedure. The same decision was applied to the genetic algorithm, which used only 10,000 images for the test (to have a fair comparison).

The table below provides the architecture of the LeNet5-inspired CNN used for the MNIST dataset experiments. This architecture was used for all training configurations:

The table below shows the results of various parameter combinations we tested during the training phase. The chosen parameters (batch size of 1024 and learning rate of $10^{-4}$) were selected because they provided the best balance between accuracy and training stability, as evidenced by the results.

Table 9 results file provides a comprehensive overview of the various parameter combinations we tested and their corresponding performance metrics. This extensive analysis underscores the importance of selecting the appropriate training parameters to achieve optimal performance in DNNs.

*Table 8.* Architecture of the LeNet5-inspired CNN for MNIST dataset experiments ($T$ stands as the batch size).

| LAYER TYPE | FILTERS | OUTPUT DIMENSION |
|---|---|---|
| CONV2D | [1x1x64] | $T$x28x28x64 |
| ACTIVAT (RELU) | - | $T$x28x28x64 |
| BATCH NORMALIZATION | - | $T$x28x28x64 |
| CONV2D | [1x1x32] | $T$x28x28x32 |
| ACTIVAT (RELU) | - | $T$x28x28x32 |
| BATCH NORMALIZATION | - | $T$x28x28x32 |
| CONV2D | [1x1x16] | $T$x28x28x16 |
| ACTIVAT (RELU) | - | $T$x28x28x16 |
| BATCH NORMALIZATION | - | $T$x28x28x16 |
| CONV2D | [3x3x8] | $T$x26x26x8 |
| ACTIVAT (RELU) | - | $T$x26x26x8 |
| BATCH NORMALIZATION | - | $T$x26x26x8 |
| CONV2D | [3x3x4] | $T$x24x24x4 |
| ACTIVAT (RELU) | - | $T$x24x24x4 |
| BATCH NORMALIZATION | - | $T$x24x24x4 |
| FLATTEN | - | $T$x2304 |
| FC | [2304x128] | $T$x128 |
| BATCH NORMALIZATION | - | $T$x128 |
| DROPOUT (0.2) | - | $T$x128 |
| FC | [128x64] | $T$x64 |
| BATCH NORMALIZATION | - | $T$x64 |
| DROPOUT (0.2) | - | $T$x64 |
| FC | [64x10] | $T$x10 |

*Table 9.* Performance of Various Training Configurations over 5,000 Epochs.

| TRAINING PARAMETERS | LEARNING RATE :1E-3 | LEARNING RATE :5E-4 | LEARNING RATE :1E-4 | LEARNING RATE :5E-5 | LEARNING RATE :1E-5 |
|---|---|---|---|---|---|
| BATCH SIZE: 256 | 99.19 | 99.19 | 99.19 | 99.2 | 99.21 |
| BATCH SIZE: 512 | 99.2 | 99.21 | 99.22 | 99.22 | 99.22 |
| BATCH SIZE: 1024 | 99.23 | 99.24 | 99.25 | 99.24 | 99.24 |
| BATCH SIZE: 2048 | 99.23 | 99.23 | 99.24 | 99.24 | 99.24 |
| BATCH SIZE: 4096 | 99.24 | 99.24 | 99.24 | 99.24 | 99.24 |

# C. Appendix-Computational Complexity Analysis

## C.1. Computational Complexity of Training a Convolutional Neural Network (CNN)

### C.1.1. NETWORK ARCHITECTURE AND TRAINING SPECIFICATIONS

The computational complexity for training a Convolutional Neural Network (CNN) is analyzed based on the following specifications:

- **Input image size**: $28 \times 28 \times 1$

- **Number of layers**: 8

  - **First 5 layers are convolutional** with kernel sizes:
    * $1 \times 1 \times 64$
    * $1 \times 1 \times 32$
    * $1 \times 1 \times 16$
    * $3 \times 3 \times 8$
    * $3 \times 3 \times 4$

  - **Last 3 layers are fully connected** with sizes:
    * 128
    * 64
    * 10

- **Batch size**: 1024

- **Training set size**: 50000

- **Number of epochs**: 5000

### C.1.2. COMPLEXITY CALCULATION FOR CONVOLUTIONAL LAYERS

The number of operations for each convolutional layer is calculated as:

$$O(K \times (W - F + 1) \times (H - F + 1) \times F^2 \times D)$$

Where:

- $K$ is the number of filters (kernels) in the layer.

- $W$ is the width of the input to the layer.

- $F$ is the spatial dimension (width and height) of the filter.

- $H$ is the height of the input to the layer.

- $D$ is the depth (number of channels) of the input to the layer.

For each convolutional layer:

- **Layer 1**: $1 \times 1 \times 64$
  - Output size: $28 \times 28 \times 64$
  - Number of operations: 50176

- **Layer 2**: $1 \times 1 \times 32$
  - Output size: $28 \times 28 \times 32$
  - Number of operations: 1605632

- **Layer 3**: $1 \times 1 \times 16$
  - Output size: $28 \times 28 \times 16$
  - Number of operations: 401408

- **Layer 4**: $3 \times 3 \times 8$
  - Output size: $26 \times 26 \times 8$
  - Number of operations: 778752

- **Layer 5**: $3 \times 3 \times 4$
  - Output size: $24 \times 24 \times 4$
  - Number of operations: 165888

### C.1.3. COMPLEXITY CALCULATION FOR FULLY CONNECTED LAYERS

The number of operations for each fully connected layer is calculated as:

$$O(N \times M)$$

Where $N$ and $M$ stand for the input size dimension and output dimension, respectively. For each fully connected layer:

- **Layer 6**: 128 neurons
  - Number of operations: 294912

- **Layer 7**: 64 neurons
  - Number of operations: 8192

- **Layer 8**: 10 neurons
  - Number of operations: 640

C.1.4. TOTAL COMPLEXITY FOR ONE FORWARD PASS

Summing the operations for all layers gives the total complexity for one forward pass:

Total forward pass operations $= 50,176+1,605,632+401,408+778,752+165,888+294,912+8,192+640 = 3,337,600$

C.1.5. TOTAL COMPLEXITY FOR ONE BACKWARD PASS

The backward pass typically requires approximately twice the number of operations as the forward pass (Goodfellow et al., 2016; nie):
$$\text{Total backward pass operations} = 2 \times 3,337,600 = 6,675,200$$

C.1.6. TOTAL COMPLEXITY FOR ONE EPOCH

To calculate the total complexity for one epoch, we first determine the number of iterations, which is the number of training examples divided by the batch size:

$$\text{Number of iterations per epoch} = \frac{\text{Training set size}}{\text{Batch size}} = \frac{50,000}{1024} \approx 48.83$$

The total complexity for one epoch is then the number of iterations multiplied by the batch size and the complexity of both forward and backward passes:

Total operations per epoch $=$ Number of iterations $\times$ Batch size $\times$ (Forward pass complexity $+$ Backward pass complexity)

$$= 48.83 \times 1024 \times (3,337,600 + 6,675,200)$$
$$= 48.83 \times 1024 \times 10,012,800$$
$$= 48.83 \times 10,250,675,200$$
$$\approx 500,000,000,000$$

C.1.7. TOTAL COMPLEXITY FOR 5000 EPOCHS

$$\text{Total operations for 5000 epochs} = 5000 \times 500,000,000,000 = 2.5 \times 10^{15}$$

**C.2. Computational Complexity of a Genetic Algorithm**

C.2.1. GENETIC ALGORITHM SPECIFICATIONS

- **Population size (P)**: 50

- **Number of generations (G)**: 50

- **Input size (N)**: 8

- **Fitness function complexity**: Equivalent to the forward pass of the CNN for a batch size of 1024

- **CNN Forward Pass Complexity for 1024 images**: $O(1024 \times 3,337,600) = O(3.417 \times 10^9)$ operations

C.2.2. COMPLEXITY CALCULATION

**Fitness Function Evaluation**    Given $F = O(3.417 \times 10^9)$, the complexity for evaluating the fitness of the entire population is:
$$O(P \times F) = O(50 \times 3.417 \times 10^9) = O(1.7085 \times 10^{11})$$

**Selection**    Assuming a selection process with complexity $O(P \log P)$:

$$O(50 \log 50) \approx O(50 \times 3.91) = O(195.5)$$

**Crossover and Mutation**   The complexity for crossover and mutation per individual is:

$$O(P \times N) = O(50 \times 8) = O(400)$$

**Total Complexity for One Generation**   Combining the fitness evaluation, selection, crossover, and mutation complexities:

$$O(P \times F + P \log P + P \times N) = O(1.7085 \times 10^{11} + 195.5 + 400) \approx O(1.7085 \times 10^{11})$$

**Total Complexity for All Generations**   The total complexity for $G$ generations:

$$O(G \times (P \times F + P \log P + P \times N)) = O(50 \times 1.7085 \times 10^{11}) = O(8.5425 \times 10^{12})$$

**C.3. Comparison**

- **Total complexity for training the CNN over 5000 epochs**: $O(2.5 \times 10^{15})$ operations.

- **Total complexity for running the genetic algorithm for 50 generations**: $O(8.5425 \times 10^{12})$ operations.

C.3.1. COMPLEXITY OF ONE EPOCH VS ONE GENERATION

- **Total complexity for one epoch of CNN training**: $O(500,000,000,000)$ operations.

- **Total complexity for one generation of the genetic algorithm**: $O(1.7085 \times 10^{11})$ operations.

C.3.2. SUMMARY

The computational complexity of training the CNN for 5000 epochs is significantly higher than running the genetic algorithm for 50 generations. Specifically, the CNN training requires approximately two orders of magnitude more operations than the genetic algorithm. Furthermore, the complexity of one epoch of CNN training is substantially higher than one generation of the genetic algorithm, emphasizing the computational expense associated with deep learning models compared to heuristic optimization methods.

