# OpenReview forum: "An Analytical Approach to Enhancing DNN Efficiency and Accuracy Using Approximate Multiplication"
_ICML.cc/2024/Workshop/WANT — WANT@ICML 2024 Poster_

### Official Review · Reviewer_kFCP · 2024-06-11

**Confidence:** 4

**Summary:**

The paper uses information theory and genetic algorithm to optimize convolutional neural networks using approximate multiplication, with emphasis a family called signed-carry-disregard-multiplier. When optimizing LeNet (MNIST dataset), they achieve a small accuracy gain while also reducing the estimated energy consumption (assuming the hardware will be implemented).

**Strengths:**

* Relevant to the workshop: has to potential to perform energy efficient training (It is not clear to me if the implementation was applied during the back-propagation phase, but it should at least make the forward more efficient)

* Very novel: It's been a while since I saw practical usage for information theory, genetic algorithms, or novel efficient multipliers--- Never-mind a method that combines all three elements.

* Interesting results: Although the gain is small, the method improves both accuracy and efficiency, which is rare.

**Weaknesses:**

* Limited experiment: Only applied to a very small network, with limited effect on the accuracy (0.16%) which is considered within the margin of error of neural networks.

* Consequently, I think that some of the strong claims made by the paper (e.g., "achieve accuracy levels previously considered unattainable"), are premature, especially when they are presented as a result of using approximate-multipliers. These statements lack theoretical justifications, since approximate-multipliers aren't expected to be better than exact multipliers.

---

### Official Review · Reviewer_E4tF · 2024-06-12
**Good analysis on how to enhance DNN efficiency with approximate multipliers**

**Confidence:** 3

**Summary:**

This paper uses the information plane to track the effect of applying approximate multipliers which simplifies the hardware components to achieve higher efficiency.

**Strengths:**

Good motivation and reasonable hypothesis. The authors attempt to use information bottleneck theory to analyze the effect of approximate multiplier which is visually informative over the model structure. It is intuitive that using approximation in LeNet improves the generalization of the models, considering the success of dropout for those models.

**Weaknesses:**

As the authors proposed, a wider study of different models and datasets could enhance the conclusion of this paper. There is no theoretical analysis of how the approximate multipliers would affect the information flow, except the fitting and compression phases can be observed empirically. Even though the authors' hypothesis matches the general intuition, the evidence in a single case is not enough to reach a strong conclusion. If it is difficult to quantify the influence of applying an approximate multiplier, a more general case study would make the conclusion more convincing.

How is energy saving defined? In Appendix A.4 some approximate multipliers achieve up to 150% energy saving which is very confusing to me. It's also unclear whether they are measured on real machines or from simulations.

---

### Meta-Review · Area_Chair_zCgS · 2024-06-18

**Recommendation:** Accept (Poster)
**Confidence:** 4

**Metareview:**

The overall reviewer sentiment for this submission appears to be positive. The reviewers appreciated the work's novelty and relevance/scope. Experimental results appear to be reasonable, with a slight accuracy improvement on LeNet/MNIST. The main shortcoming of the work, as pointed out by multiple reviewers, is the limited evaluation: a wider study involving bigger networks and more datasets is required to fully understand how well the method generalizes.

Given the relevance and novelty of the work, I'm inclined to give this submission an accept (poster).

---

### Decision · Program_Chairs · 2024-06-18

**Decision:**

Accept (Poster)

**Comment:**

We thank the authors for their time and contribution to WANT and we are pleased to share that after the reviewing process the paper has been accepted. Congratulations! We encourage the authors to consider reviewers' feedback for the improvement of the camera-ready version. We hope to see you in person at the workshop and brainstorm on efficient training research together!